# Immunisation against COVID-19 in Pregnancy and of Women Planning Pregnancy

**DOI:** 10.3390/v15030621

**Published:** 2023-02-24

**Authors:** Justin C. Konje, Mariam Al Beloushi, Badreldeen Ahmed

**Affiliations:** 1Feto-Maternal Centre Al Markhiya, Doha P.O. Box 34181, Qatar; 2Obstetrics and Gynecology Department, Weill Cornell Medicine Qatar, Doha P.O. Box 24144, Qatar; 3Obstetrics and Gynaecology, Department of Health Sciences, University of Leicester, Leicester LE2 7LX, UK; 4Women’s Wellness and Research Centre, Hamad Medical Corporation, Doha P.O. Box 3050, Qatar; 5Department of Obstetrics and Gynaecology, Qatar University, Doha P.O. Box 2713, Qatar

**Keywords:** severe acute respiratory syndrome (SARS) Coronavirus-2 (SARS-CoV-2), COVID-19, vaccination, reproduction, pregnancy

## Abstract

Following reports of the first human SARS-CoV2 infection in December 2019 from Wuhan Province, China, there was such rapid spread that by March 2021, the World Health Organization (WHO) had declared a pandemic. Over 6.5 million people have died from this infection worldwide, although this is most likely an underestimate. Until vaccines became available, mortality and severe morbidity were costly in terms of life lost as well as the cost of supporting the severely and acutely ill. Vaccination changed the landscape, and following worldwide adoption, life has gradually been returning to normal. The speed of production of the vaccines was unprecedented and undoubtedly ushered in a new era in the science of fighting infections. The developed vaccines were on the already known platforms for vaccine delivery: inactivated virus, virus vector, virus-like particles (VLP) subunit, DNA and mRNA. The mRNA platform was used for the first time to deliver vaccines to humans. An understanding of these platforms and the pros and cons of each are important for clinicians who are often challenged by the recipients on the advantages and risks of these vaccines. These vaccines have so far and reassuringly been shown to be safe in reproduction (with no effect on gametes) and pregnancy (not associated with congenital malformations). However, safety remains paramount and continuing vigilance is critical, especially against rare fatal complications such as vaccine-induced thrombocytopenia and myocarditis. Finally, the waning immunity months after vaccination means repeated immunisation is likely to be ongoing, but just how often and how many such revaccinations should be recommended remains uncertain. Research into other vaccines and alternate delivery methods should continue as this infection is likely to be around for a long time.

## 1. Introduction

Coronaviruses cause various infections in avians and mammals. They specifically attack the respiratory system causing damage, which in some cases can cause death [1]. Three new coronaviruses related severe outbreaks of zoonotic origin have occurred in the last two decades—the Severe Acute Respiratory Syndrome Coronavirus-1 (SARS-CoV-1) in 2002–2003, the Middle East Respiratory Syndrome Coronavirus (MERS-CoV) in 2012 and recently the Severe Acute Respiratory Syndrome Coronavirus-2 (SARS-CoV-2) in 2019, which was declared a pandemic by the WHO in March 2020 [2] and has been the most devastating. As of September 2022, more than 600 million cases have been confirmed, with over 6.5 million deaths, giving a mortality rate of 1.18% [3], but this is thought to be an underestimation as testing and reporting of deaths have not been universal, and furthermore, data on asymptomatic cases are often not captured.

The Severe Acute Respiratory Syndrome Coronoavirus-2 (SARS-CoV-2) is a novel enveloped ribonucleic acid (RNA) betacoronavirus belonging to the subfamily Coronavirinae in the Coronoviridae family [4,5,6]. Within this family of viruses are four genera—Alphacoronoavirus, Betacoronavirus, Gammacoronavirus and Deltacoronavirus [7]. Other members of the family that have caused infections in humans with varying severity include SARS-CoV-1 and MERS-CoV.

SARS-CoV-2 is spherical in shape with a core shell and surface with proteins, some of which project like the spikes of a crown, hence named corona (crown in Latin). The proteins are the spike (S) which is responsible for attachment to the host cell membrane receptor, followed by fusion and finally entry into the host cell, the membrane (M), which is the most abundant, the envelope (E) is characterised by a comparatively higher basic reproductive number (Ro) [*Ro is referred to as the reproductive number. It is used to gauge how infectious a contagious infection is. The basic reproductive number (Ro) of COVID-19 has been initially estimated by the WHO to range between 1.4 and 2.5]* which explains its rapid spread worldwide, and the N protein, which makes part of the helical nucleocapsid that includes the genome RNA [8]. The virus has a single-stranded positive-sense RNA of around 30 kb that is non-segmented. Phylogenetic analysis has shown that SARS-CoV-2 has an 80% genetic sequence similarity to that of SARS-CoV-1 and a 96.2% similarity to that of the bat coronavirus RaTG13 [9]. SARS-CoV-2 has a higher reproductive number (Ro) than SARS-CoV-1, implying that it spreads more efficiently [10].

Although the total number of people infected with SARS-CoV-2 worldwide continues to increase, the rate of increase has probably peaked and may indeed be falling. Pregnant women, by virtue of their relatively immunosuppressed status, tend to develop severe consequences from infections that ordinarily will not pose problems in the non-pregnant, and indeed have been shown to have a tendency to develop severe complications with SARS-CoV-2 [11]. Like most viral infections, effective treatment has been elusive, and the only approach that has been adopted globally to reduce the spread and burden of SARS-CoV-2 infection is immunisation. Concordant with the wide adoption of immunisation against SARS-CoV-2 are concerns about safety in pregnancy and around reproduction.

## 2. COVID-19 Vaccination

The severe health (morbidity and mortality), social and economic consequences of the Coronaviruses Infectious Diseases-2019 (COVID-19) pandemic led to the development of vaccines with unparalleled speed. These developments were across all six platforms for vaccine delivery (i.e., inactivated virus vaccines, live attenuated virus vaccines, recombinant viral vector vaccines, recombinant subunit vaccines, DNA vaccines and RNA vaccines) [12]. As of early 2022, at least seven different vaccines across three of these platforms had been approved in several countries. The most commonly offered ones are the Pfizer-BioNTech vaccine (mRNA), Moderna vaccine (mRNA), the Johnson & Johnson’s Janssen Ad26.COV2.S vaccine (adenoviral vector), the AstraZeneca, University of Oxford and Serum Institute of India ChAsOx1 CoV-19 vaccine (adenoviral vector-based), the Chinese Sinovac Research and Development CoronaVac and the Russian Sputnik V (Gam-COVID-Vac), an adenovirus vector vaccine. For most of these vaccines, phase II and III clinical trials did not include pregnant women, and vaccine phobia plus negative social media reports significantly affected uptake in general and more so in vulnerable groups, which include pregnant women.

The first of the approved vaccines is a messenger RNA (mRNA) vaccine made by in-vitro transcription of a target gene from a linearised DNA template [12]. Vaccines of this type, mRNA, are non-infectious and non-integrating. Hence, they are unable to cause infectious or insertional mutagenesis. They are also easily degraded, with a half-life that can be regulated through modification to the template or delivery methods [13]. Antigen-specific cellular and humoral immune responses are enhanced by the strong immune-stimulatory effect and intrinsic adjuvant activity of the in-vitro transcribed mRNA [14,15,16,17]. Other advantages of this platform include no anti-vector response (it is the minimal genetic vector), large amounts of antigen can be produced from a minimal vaccine dose since mRNA can be replicated intracellularly, and mRNA is relatively easy to produce compared to recombinant protein and live attenuated virus vaccines. Furthermore, the manufacturing process for these types of vaccines can be standardised to produce many encoded protein immunogens, making mRNA the preferred platform for rapid response during a pandemic [12]. However, some mRNA vaccines may induce potent type I interferon reactions, especially in those at an increased risk of autoimmune response. Furthermore, the lipid nano peptide (NLP) used for packaging the mRNA means the extracellular RNA they deliver can induce thrombosis [12]. Indeed, vaccine-induced thrombosis (VIT) has been reported as a complication in a small number of cases [18]. In pregnancy, this risk may be increased because of the thrombotic state induced by the physiological changes of pregnancy [1]. The Pfizer-BioNtech and Moderna COVID-19 vaccines are lipid nanoparticle-formulated mRNA vaccines which do not contain ingredients that are known to be harmful to pregnant women or to the fetus. These vaccines do not enter the nucleus and do not alter human DNA in recipients. As a result, the mRNA in the vaccine is translated and transcribed by the body to produce the spike protein, which then acts as an intracellular antigen to stimulate the immune response. The mRNA in the vaccine is normally degraded within a few days and cannot be incorporated into the host genome. Thus, it cannot cause any genetic changes [12].

The second platform for COVID-19 vaccines that received approval for use is the viral vector platform. The vaccines of this platform are the Johnson and Jonson-Janssen, Astra-Zeneca and Russian Sputnik vaccines [19]. When attenuated viruses are used as platforms for delivering vaccines for highly infectious pathogens such as SARS-CoV-2, Ebola virus and Zika virus, there is the danger that these viruses may revert to their pathogenic forms and cause disease (i.e., recover their virulence) [20]. Therefore, recombinant viral vectors are developed that mimic the natural pathogen but are not virulent. This is achieved by cloning the target pathogen into an avirulent viral host—in this case, an adenovirus. The adenovirus (host-virus genome) carries out all the viral transmission and amplification functions, including antigen production, to present the chosen antigen to the patient’s immune system [12]. The vaccines in this category introduce DNA for the SARS-CoV-2 spike protein into the cells. The DNA is first copied into mRNA, which is then used to produce copies of the spike protein that stimulates the body’s immune system to generate antibodies. The main advantage of this platform is the fact that since it presents protective antigen epitopes in the context of a live, replicating virus without concern about reversion to a pathogenic state, it induces a more robust immune response and allows for antigen sparing as the viral vectors have limited nucleotide activity [12]. A disadvantage of this platform is that the immunogenicity induced resulting from both the encoded antigen and the viral vector antigen may lower or modify the vaccine’s efficacy by diverting immune responses away from the target antigens [12]. In the unlikely event that an individual has pre-existing immunity to the viral vector, it can lead to premature clearance of the vaccine before an immune response can be mounted to the antigen of interest carried by the vector making it ineffective [21,22].

The third platform that COVID-19 vaccines belong to is the antigenic sub-unit, with examples in this group including Ahnui Zhifei Longcom Biopharmaceutical Novovax, GSK Sanofi Pasteur and United Biomedical COVAXX vaccines. These do not contain any live component of the pathogen (SARs-CoV-2) but contain an antigenic sub-unit of the pathogen that induces the protective immune response [23]. Vaccines on this platform thus include only parts of the virus or bacteria necessary to cause a protective immune response; hence side effects are less common (local ones are negligible) since the sub-units do not contain other molecules of the pathogen [23]. Another advantage of these groups of vaccines is that they are less likely to induce an eosinophilic immunopathology or antibody-mediated disease enhancement, hence are safe for use in those who are immunosuppressed [24].

A summary of the information on the four vaccines with published outcomes following use in pregnancy is shown in Table 1. The mRNA vaccines have been shown to have a higher efficacy in the prevention of symptomatic illness with efficacies of about 95% compared to that of about 70% with adenoviruses vaccines. With regard to severe COVID-19, there are no obvious differences in efficacies.

## 3. Immunological Response to Vaccination in Pregnant Woman

Of the four vaccines that have been used extensively, three are administered as two or more vaccine doses three to twelve weeks apart, while the 4th, the Johnson and Johnson-Janssen vaccine, is administered once. Consideration should, however, also be given for a booster dose about six months later for all of them. While most studies on the immune response have been on non-pregnant women, Prabhu et al. [36], Collier and McMahan [37], and Gray et al. [38] have all shown that the immunological response in pregnant women from measured IgG increase is similar to that in non-pregnant women. Interestingly IgG has been measured in the cord blood and breastmilk of mothers who were vaccinated during pregnancy in New York [36]. In the New York study, it was shown that antibody production (IgG) started five days after immunization. By day 16, the antibody production was detectable in cord blood. The implication of these findings is that if mothers are vaccinated at least 16 days prior to delivery, the baby should be born with passive immunity, which would then be supplemented with the antibodies secreted in breast milk if they breastfeed. Children of mothers vaccinated after 20 weeks of gestation have been shown to have a lower risk of hospitalisation compared to those of mothers vaccinated before 20 weeks (85% vs. 32%). These results suggest a declining passive immunity as the antibody levels in the mother wane [39]. Several studies have shown a decline in antibodies by 4–6 months after the booster dose, and following these studies, 3rd and 4th booster doses have been recommended, especially for vulnerable groups. It would therefore seem logical to consider a 3rd dose in women who are vaccinated prior to pregnancy and have the second booster dose early in pregnancy or, indeed, prior to pregnancy. Table 2 is a summary of the data from some of the studies on the immunogenicity of COVID-19 vaccination in pregnancy. The data from these studies suggest that the early third trimester may be the optimal time for a booster vaccine dose (supporting our recommendation above) to allow for optimal maternal-to-fetal antibody transfer for appropriate neonatal immunity. While the transplacental transfer of antibodies starts in the second trimester, it is most efficient in the third trimester [40].

## 4. COVID-19 Vaccination and Pregnancy

The effectiveness of vaccines in reducing risk of moderate to severe illness, hospitalisation, and deaths from COVID-19 has been shown to be extremely high (69–95%) [1,59,60,61,62], although immunity has been demonstrated to wane 4–6 months after booster doses [63]. Furthermore, the regular emergence of new/mutated variants has been regarded as a potential factor affecting vaccine efficacy [64,65].

Concerns about safety with the rapidity with which these vaccines were developed have, in some cases, resulted in vaccine hesitance, especially in vulnerable groups such as pregnant women and others at risk. Several studies, most of which are cohort studies, have investigated adverse pregnancy outcomes in women following COVID-19 vaccination (Table 3) [42,44,45,50,53,66,67,68,69,70,71,72,73,74,75,76,77,78,79,80,81,82]. The best evidence for the safety of COVID-19 vaccination in pregnancy comes from the USA Centers for Disease Control and Prevention (CDC) tracking of mRNA vaccine recipients who were either pregnant at the time of vaccination (V-safe) or shortly after. In the latest report published on 35,691 women v-safe participants aged 16 to 54 years identified as pregnant who received the Pfizer-Biotech Vaccine, injection-site pain was reported more frequently among pregnant persons than among non-pregnant women, whereas headache, myalgia, chills, and fever were reported less frequently. Of those who completed their pregnancies, 115 (13.9%) were pregnancy losses, and 712 (86.1%) were live births. Adverse neonatal outcomes included preterm birth (9.4%) and small size for gestational age (3.2%); no neonatal deaths were reported. Although not directly comparable, calculated proportions of adverse pregnancy and neonatal outcomes in persons vaccinated against COVID-19 who had a completed pregnancy were similar to incidences reported in studies involving pregnant women that were delivered before the COVID-19 pandemic [70]. From all these studies, we are able to conclude that there have so far been no safety concerns in pregnant women who received especially mRNA and, to a lesser extent, the adenovirus COVID-19 vaccines. However, more longitudinal follow-up, including follow-up of large numbers of women vaccinated early in pregnancy, is necessary to inform maternal, pregnancy, and infant outcomes [1,70]. These follow-up studies are continuing, and hopefully, some will be on the adenovirus platform vaccines. Overall, the evidence from these supports the conclusion that the risks of COVID-19 in the un-vaccinated outweigh those of the vaccines.

## 5. COVID-19 Vaccination and Transplacental Transfer of Antibodies

Maternal antibodies (IgG) produced in response to infections cross the placenta and provide passive immunity to the baby when exposed to the infective pathogen after birth. Some vaccines, when administered to pregnant women, offer protection to both the mother and infant as the induced antibodies also cross the placenta. Examples include the TDap vaccine (against acellular pertussis, diphtheria and tetanus toxoids) [83,84]. With regards to COVID-19 vaccines, there is evidence that generated antibodies from vaccinated mothers reach their fetuses transplacentally. In a study evaluating SARS-COV-2 antibody titre in cord blood from 16 neonates whose mothers received the SARS-COV-2 mRNA vaccine (BNT162b2), all maternal and cord blood samples were positive for SARS-CoV-2 spike (S) protein antibody. The anti-S antibody titre in the cord blood increased the longer the interval (weeks) between the first dose of the vaccine and delivery [47]. In another study, there was a positive correlation between the number of weeks from the first or second vaccine and the ratio of the umbilical cord to the maternal SARS-CoV-2 anti-S antibody [85]. Mithal et al. studied 27 vaccinated pregnant women (23 had received the BNT162b2/Pfizer or Spikevax (Moderna) vaccine, four had received an unknown SARS-CoV-2 vaccine) and their infants (28 including a pair of twins). Of these women, 74% received both doses before delivery. IgG against SARS-CoV-2 was present in all but three of the infants whose mothers had received their first dose of vaccine less than three weeks before delivery. They concluded that “receiving both vaccine doses before delivery and longer latency from vaccination to delivery was associated with a higher IgG concentration in infants and stronger immunity” [52]. This result was also confirmed in other studies [86,87]. Prabhu et al. studied 122 pregnant women, of whom 55 and 67 had received their first and second mRNA vaccine doses, respectively, before delivery. They showed that maternal antibody production started five days after the first vaccine dose, and transplacental transfer to the fetus began after the 16th day post-vaccination. These findings emphasise the importance of a second vaccination dose, with the data showing that 99% of women receiving both doses had IgG antibodies in their cord blood samples; however, only 44% of women who had received one dose of the vaccine had cord blood IgG antibodies against SARS-CoV-2. They concluded from this that transplacental antibody transfer rises with the increase in weeks between the second vaccine and birth [36]. It would seem from these data that an interval of four weeks between maternal vaccination and delivery is necessary to enable adequate antibodies to be generated and transferred to the fetus for the protection of the infant against SARS-CoV-2. Since adequate antibody production is achieved two weeks after the second dose of the vaccine (bearing in mind that there usually are at least 21 or 28 days between first and second doses with different vaccines) and that transplacental antibody transfer begins approximately at 16 weeks of gestation, vaccinating pregnant women at the beginning of the second trimester might lead to the highest levels of antibodies being transferred to the newborn [88]. Comparisons of vaccination in early and late third trimesters showed that early vaccination increases antibody transfer through the placenta and increases neonatal neutralizing antibody levels [89]. With regards to the transplacental antibody transfer, the highest transfer occurred in women vaccinated in the first followed by the second trimester; antibody titres declined more when the vaccination was in the first trimester. This would imply that a booster vaccine in the third trimester would be of maximum benefit to the fetus [90]. Data on the protective effect of vaccine-induced antibodies against SARS-COV-2 are limited. In their study, Atyeo et al. [38] did show that the IgG antibody levels against the S protein in maternal and cord samples were not statistically different although cord levels were lower.

## 6. COVID-19 Vaccination and Breastfeeding

Antibodies against SARS-CoV-2 are present in breast milk following maternal infection and offer passive immunity to the newborn. The same will apply if vaccine-induced antibodies are also present in breastmilk. In a study of fourteen lactating women who received two doses of the BNT162b2 (Pfizer) vaccine, it was found that 3–7 days after the second dose of the vaccine, there was a peak in the amount of SARS-CoV-2 IgA and IgG antibodies in breastmilk. IgG remained stable until 4–6 weeks, but IgA levels fell significantly. There was also a minimal transfer of vaccine mRNA into breastmilk. Breastfed infants showed no adverse effects during the first 28 days of follow-up [91]. These findings were further confirmed by another study which found IgA and IgG antibodies in breastmilk as early as two and four weeks after the first dose of the vaccine, respectively [92]. Studies of post-vaccination lactating women who received the mRNA vaccines similarly found SARS-CoV-2 IgA and IgG antibodies in breastmilk samples that were absent before vaccination [93]. A recent systematic review concluded that after the first vaccine, 64% and 30% of breastmilk samples were positive for IgA and IgG, respectively, and these rose to 70% and 91% after the second dose [94]. Evidence on the ability of these antibodies to neutralise antigens is limited. In an in-vitro study of 34 cases, the neutralizing capacity of antibodies in milk samples obtained pre- and post-SARS-COV-2 infection (with IgA and IgG antibodies) were compared; 62% of the post-infection samples compared to none of the pre-infection samples were able to neutralise antigen suggesting that these antibodies are capable of providing passive immunity [95]. With regards to vaccine-generated antibodies, a study by Young et al. showed that 60% and 85% of human milk samples had neutralizing capacity against the wild-type SARS-CoV-2 virus after the first and second vaccine doses, respectively; of the post-infection samples, 80% and 100% showed neutralizing capacity after 28 and 90 days, respectively. Perez et al. [96] showed that the neutralizing capacity of breastmilk three and six months post-vaccination was 83.3%, 70.4% and 25%, respectively and correlated strongly with IgG levels in the milk [97]. Taken together, we can conclude that vaccinated-generated antibodies cross to the baby through breastmilk and that these antibodies are able to offer passive immunity to the baby. These additional benefits of vaccination should form part of the counselling of pregnant women.

## 7. Vaccination and Reproduction

SARS-CoV-2, through its spike (S) protein, binds to the angiotensin-converting enzyme 2 (ACE-2) receptor [98] after it enters the body. The interaction between the virus and the receptor is mediated by the transmembrane protease serine 2 (TMPRSS2) or cathepsins B and L (encoded by the genes CTSB and CTSL, respectively) when TMPRSS2 is absent [99]. TMPRSS2 and ACE-2 receptors [100] have been localised in parts of the female genital tract, such as the ovaries, where ACE-2 receptor activity is expressed in relation to folliculogenesis and maturation, steroid synthesis and ovulation. The distribution of these receptors is, at best, described as sparse. Because of their presence, therefore these parts of the female reproductive tract are a potential target for SARS-CoV-2. There is the potential for the SARs virus to cause damage via the ACE2 /TMPRSS2 pathway [101,102]. In males, ACE-2 receptors and TMPRSS2 (targets of the virus) are expressed on testicular tissue, again implying a potential of damage by SARS-CoV-2 to the testis and possibly having an adverse effect on its function [103,104]. Furthermore, testosterone has been shown to facilitate SARS-CoV-2 spread via activation of TMPRSS2 [105], thus increasing the attractiveness of the virus onto the testicular tissue. SARS-CoV-2, through binding to the ACE2 receptor, may potentially cause orchitis and possible testicular atrophy and sub-fertility [106]. Furthermore, there have been reports of isolation of SARS-CoV-2 in the semen of infected males [92], raising the possibility of sexual transmission. Recent studies have shown that COVID-19 (even moderate disease) causes adverse changes in semen parameters, but these were found to recover quickly after the infection [107,108].

Clinical trials with COVID-19 vaccines neither included pregnant women nor those undergoing fertility treatment or, indeed, those in early pregnancy. Hence the impact/effect of these vaccines on gametes and early embryos is unknown. In a joint statement, however, the International Federation of Fertility Societies (IFFS) and the European Society of Human Reproduction and Embryology (ESHRE) advised that women planning to conceive in an environment with ineffective control of COVID-19 and/or with limited resources for vaccination should adopt measures to mitigate the risk of exposure and defer pregnancy pending improvement with regards to SARS-CoV-2 infection or until vaccination is available, thus indicating that vaccination is not contra-indicated in these women [109,110,111,112,113,114,115]. In a recent prospective study of the effect of the COVID-19 vaccine on sperm parameters, Gonzalez et al. [116] not only show that there were no adverse effects with the mRNA vaccine but that there seems to be an improvement in parameters in oligozoospermic men. Furthermore, the vaccine has been shown to have no effect on response to ovulation stimulation as well as parameters of successful stimulation [117,118,119], and in animals, no adverse effects on reproduction have been reported [27,120].

## 8. Complications of COVID-19 Vaccines

Side effects of COVID-19 vaccines can be localised or systemic. The most commonly-reported ones are headache, fatigue, muscle and joint pain, fever and chills and pain at the site of injection [121]. From the CDC’s Vaccine Adverse Events Reporting System (VAERS) uncommon side effects include anaphylaxis reported in approximately five cases per one million vaccine doses administered, thrombocytopenia syndrome (TTS) a rare but serious adverse event after the Johnson and Johnson-Janssen (J & J/Janssen) COVID-19 vaccination, reported in approximately four cases per one million doses administered, Guillain-Barre Syndrome (GBS) (a rare disorder where the body’s immune system damages nerve cells, causing muscle weakness and sometimes paralysis) reported mainly in men aged 50 years and older who received the J & J/Janssen COVID-19 vaccine and myocarditis and pericarditis mostly reported after Pfizer-BioNTech or Moderna (mRNA COVID-19 vaccines), particularly in male adolescents and young adults, the rates being highest after the second dose: 70.7 cases per one million doses of Pfizer-BioNTech in those age 12–15 years, 105.9 cases per one million doses of Pfizer-BioNTech in those aged 16–17 years and 52.4 cases and 56.3 cases per million doses of Pfizer-BioNTech and Moderna in those age 18–24 years, respectively [122,123].

Of these rare complications, the prothrombotic syndrome (vaccine-induced thrombotic thrombocytopenia (VITT) or vaccine-induced prothrombotic immune thrombocytopenia (VIPIT) has the potential to be serious in pregnancy (Johnson & Johnson-Janssen and AstraZeneca, University of Oxford) [124]. This is thought to be caused by immunoglobin (Ig) antibodies that bind to platelet factor 4 (PF4), also known as CXCL4. Platelets are activated following this binding to PF4 with the resultant stimulation of the coagulation system and the ensuing clinically significant thromboembolic complications [125]. The thrombosis caused by these vaccines includes cerebral venous thrombosis, splanchnic vein thrombosis (e.g., mesenteric vein, portal vein, splenic and hepatic vein), adrenal vein thrombosis (which may present as adrenal failure), pulmonary embolism and arterial thrombosis (including ischaemic stroke and acute limb ischaemia) [124,125]. Although a very rare complication, it is associated with mortality. Since pregnancy per se more than doubles the risk of VTE (because of the hypercoagulability state secondary to physiological changes in pregnancy) and severe SARS-CoV-2 doubles the risk of VTE in pregnant women with severe illness, it would be prudent to avoid the adenovirus vaccines in those who are planning a pregnancy or are pregnant and have risk factors for venous thromboembolism (VTE).

## 9. Recommendations

From these data and the risk-benefit analysis, various bodies and authorities, including the Centers for Disease Control and Prevention (CDC) [126], the Society for Maternal-Fetal Medicine (SMFM) [127,128], the American College of Obstetricians and Gynecologists (ACOG) [29,129], the International Federation of Gynaecology and Obstetrics (FIGO) and the Royal College of Obstetricians and Gynaecologists, on balance all recommend that COVID-19 vaccines should be offered to both pregnant and lactating mothers [130]. Several countries have also given regulatory approval for their use in pregnancy and in women planning a pregnancy or undergoing treatment for infertility. The recommendations and positions of the various societies on COVID-19 vaccination in pregnancy and reproduction are shown in Table 4. In counselling women for vaccination, particular attention should be given to those who are at high-risk. These include women from ethnic minorities, overweight/obese and/or with co-morbidities [131].

## 10. Conclusions

The COVID-19 pandemic undoubtedly changed all aspects of clinical practice and, indeed, life. In the absence of effective treatment, vaccination remains the only option for controlling the infection. The rapidity with which these vaccines were developed left several questions in the vaccine doubters and, indeed, was partly responsible for the reluctance of some to receive them. The benefits of vaccination greatly outweigh the side-effects. The available accumulated evidence since the vaccines were introduced overwhelmingly confirms their safety in all stages of pregnancy and in those planning pregnancies. It is the recommendation of most international and national societies that women planning a pregnancy or are pregnant or lactating should be offered vaccination against COVID-19. Where vaccination was received before pregnancy or in early pregnancy, consideration should be given to a booster dose in the third trimester to allow for the maximum transplacental transfer of antibodies to the baby for passive immunity afterbirth. While the mRNA vaccines are preferred by many, the J & J Janssen vaccine as well as the protein platform vaccine, are also acceptable options. We believe that while this is indeed a good practice to offer these vaccines, caution must continue to be exercised going forward, and more data must be collected and regularly reviewed, especially that which includes vulnerable groups like pregnant women, those who are immunocompromised, children and the elderly. With the levels of antibodies waning after vaccination, questions remain on how often this should be administered and whether repeated vaccinations are associated with more complications.

## Figures and Tables

**Table 1 viruses-15-00621-t001:** Authorised vaccines for use in pregnancy, including their efficacy and safety.

Vaccine	Manufacturer/Country	Platform/ Technology	No of Doses	Efficacy from Randomised Trials	Reported Adverse Effects	Reproductive Toxicology Studies
Pfizer-BioNTech mRNA (BNT16b2) [25]	USA—Pfizer	mRNA—encodes stabilised spike, lipid nanoparticle	2 doses at least 3 weeks apart	95% against symptomatic COVID-19	Myocarditis—more after 2nd dose. Reported rate is 3.5/1,000,000 after 2nd dose. Highest in age 18–29 years [26]	No reported safety concerns in rats given vaccine before and during pregnancy—investigated—effect on gonads/gametes fertility, embryo-fetal development, growth and neurofunction [27,28,29]
Moderna mRNA 1273 [30]	Moderna USA	mRNA—encodes stabilised spike, lipid nanoparticle	2 doses 4 weeks apart	94.1% against symptomatic COVID-19	Myocarditis– especially after 2nd dose—rate 3.5/1,000,000; highest in 18–29 age group [26]	No reported safety concerns in rats given vaccine before and during pregnancy—investigated—effect on gonads/gametes fertility, embryo-fetal development, growth and neurofunction [28,29,31]
Johnson and Johnson-Janssen Ad26 COV2.S [32]	Netherlands, Belgium and USA	Replication incompetent human adenovirus type 26 vector-stabilised spike	One dose	66.1% against moderate to severe-critical COVID-19; 85.4% against severe-critical COVID-9	Thrombosis with thrombocytopenia syndrome—rate 3/1,000,000 cases. Highest in females aged 30–49 years. Guillain-Barre syndrome—7.8/1,000,000 cases; highest in 50–64 age group [26]	No reported safety concerns in rats given vaccine before and during pregnancy—investigated—effect on gonads/gametes fertility, embryo-fetal development, growth and neurofunction [28,29,33]
Oxford-AstraZenecaChAdOx1 (AZS1222) [34]	AstraZeneca UK	Recombinant replication of deficient chimpanzee adenoviral vector encoding SARS-CoV-2 spike protein	Two doses 4–12 weeks apart	70.4% against asymptomatic COVID-19; 100% against severe-critical COVID-19		Development and reproductive toxicology studies have not shown harmful effects in pregnant animals and their offspring [35]

**Table 2 viruses-15-00621-t002:** Evidence of immunoreactivity from published studies in pregnancy (adapted from Badell et al. [40]).

Author, Year and Study Type	Vaccine Type	Gestational Age (Weeks) at Vaccination	Sample Size	Seropositive Maternal Blood	Seropositive Cord Blood
Shen et al., 2022 Prospective cohort [41]	Moderna	28–33	Total—29 (1st dose only 4, 1st & 2nd dose 25)	Antibody against wild type SARS-CoV-2—neutralizing IgG 40% > one dose and 97.5% > 2 doses Delta variant—neutralizing IgG 4% > 1 dose and 80.5% > 2 doses	Antibody against Wild type—Neutralizing IgG 43.3% > 1 dose and 97.4% > 2 doses Delta variant—Neutralizing IgG 1.4% >1 dose and 66.3% > 2 doses
Beharier et al., 2021 Multicentre cohort [42]	Pfizer	34	Vaccinated—86; infected—65 and control—62	Antibody against SARS-CoV-2 Rise in anti-S and anti-RBD titres > 15 days of 1st dose Additional rise after 2nd dose Vaccinated higher anti-S1 and anti-RBD but natural infection—higher anti-S2 and N antibodies	Antibody against SARS-CoV-2 Levels of anti-IgG and anti-RBD were the same after vaccination and natural infection, but transfer ratios for anti-S1, anti-S2 and anti-RBD than with natural infection
Ben-Mayor et al., 2021 Prospective multicentre [43]	Pfizer	34–37 weeks	58 (1st does 19, 2nd dose 29)	Antibody against SARS-CoV-2 Anti-IgG anti-S > 50 AU/mL detected in 53 samples	Antibody against SARS-CoV-2 Detected in 51 cord samples and negative in 7, all with a 1st dose to delivery interval of <27 days Maternal antibodies tires positively correlated with cord titres
Bookstein et al., 2021 [44]	Pfizer	Unavailable—not provided	650 (390—pregnant and 260—non-pregnant controls)	Antibody against SARS-CoV-2 Pregnant women had significantly lower serum IgG titres than non-pregnant women	No measured
Kashani-Ligumsky et al., 2021, Cohort [45]	Pfizer	No reported	29—vaccinated, 29—infected, 21—un-vaccinated controls	Antibody against SARS-CoV-2 Mean titre of 224.7 U/mL in vaccinated v 83.U/mL in infected	Antibody against SARS-CoV-2 Anti-S IgG positive in 100% of cord samples in vaccinated Mean neonatal titres significantly higher in vaccinated vs. unvaccinated (225 U/mL vs. 83.7 U/M, *p* ≤ 0.05)
Kulgelman et al., 2021, Prospective cohort [46]	Pfizer	25	130	Antibody against SARS-CoV-2 IgG antibody positive in 100% of cases Change in maternal antibody level per week increase from 2nd vaccine dose to birth—10.9%	Antibody against SARS-CoV-2 100% for IgG titre 2.6 times higher than maternal titres Change in antibody titre per week increase from 2nd vaccine doe to birth (−11.7%)
Nir et al., 2021; Prospective cohort [47]	Pfizer	33	Total—64 fully vaccinated (1st & 2nd dose); unvaccinated and recovered from COVID-19 = 11	Antibody against 100% positive for SARS-CoV-2 IgG Recovered from COVID-19 IgG-positive 26.1% in vaccinated IgG positive in 2.6% in unvaccinated	Antibody against SARS-CoV-2 98.3% in fully vaccinated 20.2% in recovered vaccinated vs. 3.3% in recovered unvaccinated
Rottenstreich A et al., 2021; Prospective cohort [48]	Pfizer	3rd trimester	20 fully vaccinated (1st & 2nd dose)	Antibody against SARS-CoV-2 anti-S and anti-RBD in 100% of cases	Antibody against SARS-CoV-2 anti-S and anti-RBD 100% Cord levels positively correlated with maternal levels Cord levels increased with increasing time from 1st vaccine to delivery
Rottenstreich A et al., 2021; Cohort [49]	Pfizer	3rd trimester (27–36)	171	Antibody against SARS-CoV-2 anti-S and anti-RBD—100% in all cases at time of delivery Titres are lower in those vaccinated in early vs. late third trimester	Antibody against SARS-CoV-2 anti-S and anti-RBD—100% in neonatal serum; higher in those whose mothers were vaccinated earlier Transfer higher after early vaccination Positive correlation of antibody titre with duration from vaccination
Rottenstreich M et al., 2022; Prospective Cohort [50]	Pfizer	1st, 2nd and 3rd trimesters	1st trimester = 90 2nd trimester = 124 3rd trimester = 188	Quantified IgG antibodies in maternal serum 1st trimester—anti-S 76 AU/mL; anti-RBD 478 AU/mL 2nd trimester—anti-S 126 AU/mL, anti-RBD-1263 AU/mL 3rd trimester—anti-S-240 AU/mL, anti-RBD-5855 Au/mL Booster in 3rd trimester of vaccine in 1st trimester—anti-S—1665 AU/mL, ani-RBD-20,956 AU/mL	Antibodies against SARS-CoV-2 positive for anti-S and anti-RBD in all neonates 1st trimester—anti-S 126 AU/mL, anti-RBD-1140 AU/mL 2nd trimester—anti-S-204 AU/mL; anti-RBD-8038 AU/mL 3rd trimester—anti-S255 AU/mL, anti-RBD-8038 AU/mL Booster in 3rd trimester after 1st trimester vaccine—anti-S-528 AU/mL, anti-RBD-4225 AU/mL
Atyeo et al., 2021 [51]	Pfizer and Moderna	16–32	64 Moderna—1st dose—32, 2nd dose—19 Pfizer—1st dose—32, 2nd dose—17 Non-pregnant controls—14	Antibody against SARS-CoV-2 Fc receptor antibody titres are significantly lower in pregnancy compared to non-pregnant > 1st dose Fc receptor antibody increase significantly > 2nd dose but is still lower than in non-pregnant women	Antibody against SARS-CoV-2 Vaccinated maternal/cord blood pairs—8, maternal titres of all antibodies higher than cord titres Enriched RBD-specific FcyR3a binding in cord
Collier et al., 2021 [36]	Pfizer and Moderna	All three trimesters	Vaccinated—103 (pregnant—30; non-pregnant—57; infected—28 of which 22—pregnant and 6 nn-pregnant0 Pfizer—11	Antibody against SARS-CoV-2 Pregnant vaccinated mean RDB IgG—27,601 vs. 37,839 for non-pregnant Mean RBD IgG of 1321in pregnant infected vs. 910 in non-pregnant infected Mean neutralizing antibody titre of 148 in pregnant vs. 901 in non-pregnant	Antibody against SARS-CoV-2 Vaccinated maternal and cord-paired samples—9; ○ Median cord RDB IgG tires higher than maternal blood titres 919,873 vs. 14,953) ○ Median cord neutralizing IgG lower h = than maternal blood (324 vs. 1016)
Mithal et al., 2021, Prospective cohort [52]	Pfizer, Moderna and unknown	33	28 total (Pfizer—18, Moderna—6, unknown—4)	Antibody against SARS-CoV-2 97% IgG positive	Antibody against SARS-CoV-2 IgG antibodies were positive in 25/28 cases 3 negative—had 1st dose < 3 weeks before delivery Increased latency from vaccination to delivery is associated with an increased transfer ratio 2nd dose before delivery is associated with increased infant IgG levels Latency from vaccination to delivery assocaity4ed with increased infant IgG levels
Shanes et al., 2021 [53]	Pfizer, Moderna and unknown	All 3rd trimester	84 vaccinated (Pfizer—49; Moderna—25, mRNA unknown—9) and 116 unvaccinated	Antibody against SARS-CoV-2 Anti-IgG level 22.8 in vaccinated and 0.04 in unvaccinated	Antibody against SARS-CoV-2 Not measured
Gray et al., Prospective cohort [54]	Pfizer and Moderna	All trimesters (1st—11, 2nd—39 and 3rd—34)	Total—100 Pregnant (Pfizer—41, Moderna—43), non-pregnant (Pfizer—8, Moderna—8)	Antibody against SARS-CoV-2 Comparable IgG levels between pregnant and non-pregnant-(i.e., no difference) All IgG titres higher in non-pregnant than pregnant with previous SARS-CoV-2 infection	Antibody against SARS-CoV-2 10 cord samples studies ○ All positive for anti-S and anti-RBD IgG ○ Neutralizing antibodies lower in cord than maternal serum—but not statistically significant
Trostle et al., 2021 [55]	Pfizer and Moderna	All trimesters (1st—2, 2nd—30, 3rd—4)	Total—36 (Pfizer—26, Moderna—10)	No measured	Antibody against SARS-CoV-2 IgG anti-S in 100% of cases in 34 cases, titre > 250 IU/mL, 2 < 250 U/mL (both vaccinated > 20 weeks before delivery)
Citu et al., 2022 [56]	Pfizer and Janssen	Third trimester	227 vaccinated Unvaccinated 608 Vaccinated without COVID-19 history—173 Unvaccinated without COVID-19 history—529 Non-pregnant—227	Antibody against SARS-CoV-2 *Prior to vaccination—N = 173 seronegative (A) vs. 54 seropositive (B) (in the pregnant cohort)* Before vaccination—0.41 vs. 145 U/mL 2 months > vaccination IgG increases to 1697 U/mL in A vs. 14,571 in B 4 months > vaccination 1083 U/mL vs. 12,571 U/mL (A vs. B), respectively) *Seronegative pregnant (N = 173) vs. non-pregnant without a history of COVID-19 (N = 173)—IgG levels* Before vaccination—0.41 vs. 0.40 U/mL 2 months > vaccination 1697 vs. 1705 U/mL 4 months > vaccination 1083 vs. 1114 U/mL	Not studied
Gloeckner et al., 2021 [57] Observational cohort	Pfizer, Moderna or AstraZeneca	21–28	Primary vaccine AstraZeneca, booster 12 weeks later with Pfizer or Moderna	Antibody against SARS-CoV-2 IgG-S antibodies detected in all samples	Antibody against SARS-CoV-2 IgG-S antibodies detected in all samples
Yang et al., 2021, Retrospective cohort [58]	Pfizer, Moderna and Janssen	All trimesters (pre-pregnancy—38, 1st trimester—193, 2nd trimester—699, 3rd trimester—429)	1359 (Pfizer—1025, Moderna—301, Janssen—33)	Antibody against SARS-CoV-2 *Anti-S IgG measured in all women* Pre-pregnancy—(Pfizer group—3.7; Moderna group—4.8 & Janssen group—NA) 1st trimester (Pfizer—3.9; Modernal—4.8; Janssen—3.6) 2nd trimester (Pfizer—4.8; Moderna—5.7; Janssen—3.0) 3rd trimester (Pfizer—6.2; Moderna—6.6; Janssen—2.5) *Anti-IgG in fully vaccinated women ≥ 14 days after 2nd dose* Pre-pregnancy (Pfizer—3.7; Moderna—4.8; Janssen—NA) 1st trimester (Pfizer—3.9; Moderna—4.8; Janssen—3.6) 2nd trimester (Pfizer—4.8; Moderna—5.7; Janssen—3.0) 3rd trimester (Pfizer—6.4; Moderna 7.1; Janssen—2.5)	Antibody against SARS-CoV-2

**Table 3 viruses-15-00621-t003:** Summary from studies that report on pregnancy outcomes after COVID-19 vaccination.

Author and Year	Type of Study and Country	Vaccine Type and Number Vaccinated	Pregnancy Outcome
Kharbanda et al., 2021 [66]	Case-controlled—USA	Vaccinated = 23,375 [Pfizer (*n* = 8267) Moderna (*n* = 15,108), Janssen (*n* = 528)] versus unvaccinated—90,338	No increase in odds of spontaneous miscarriage Spontaneous miscarriage within 28 days of vaccination—1128/13,160) 8%)
Lipkind et al., 2022 [67]	Observational retrospective—USA	Vaccinated 10,064 (Pfizer—5478; Moderna—4162; Janssen—424) Unvaccinated—36,015	Outcomes presented in vaccinated vs. unvaccinated, respectively FGR/SGA—8.2% vs. 8.2% (NS) Preterm birth—4.9% vs. 7.0% (NS; *p* = 0.06)
Morgan et al., 2022 [68]	Retrospective cohort USA	Vaccinated 1332 incompletely vaccinated or unvaccinated—8760 Pfizer (883), Moderna (382), Janssen (67)	Outcomes presented in vaccinated vs. unvaccinated, respectively Stillbirth—0% vs. 0.07% (NS) Maternal death—0% vs. 0.01% (NS)
Ruderman et al., 2022 [69]	Cohort USA	Vaccine type not defined Vaccinated in the organogenesis window—1149 Vaccinated outside the window of organogenesis—1473 Unvaccinated—534	Congenital malformations—main outcome—no difference Vaccinated 30 days before conception and up to 14 weeks—4.2% Vaccinated from 10 weeks—4.0% vaccinated outside organogenesis period—4.2% Unvaccinated—4.4%
Shanes et al., 2021 [53]	Cohort USA	Pfizer Vaccinated—84 Unvaccinated—116	Outcomes presented in vaccinated vs. unvaccinated, respectively Vaginal delivery—79% vs. 65% (*p* = 0.04) Decidual arteriopathy (NS) Fetal vascular malperfusion (NS) Low or high-grade villitis (NS)
Shimabukuro et al., 2021 [70]	V-safe Surveillance System and VAERS Registry—USA	Vaccinated (Pfizer 1st dose—9052; 2nd dose—6638; Moderna—1st dose—7930, 2nd dose—5635)	Outcomes not different from unvaccinated uninfected population Spontaneous miscarriage—12.6% Major congenital malformations—2.2% Preterm birth—9.6% SGA/FGR—3.2%
Theiler et al., 2021 [71]	Cohort—USA	Pfizer—127; Moderna—12, Janssen—1	Outcomes presented in vaccinated vs. unvaccinated, respectively FGR/SGA—7.9% vs. 6.5% (NS) Any antenatal complication—5% vs. 4.9% (NS) Stillbirth—0.0% vs. 0.3% (NS) Cesarean section—31.4% vs. 29.8% (NS) NICU admission—0.07% vs. 0.6% (NS)
Trostle et al., 2021 [72]	Case series—USA	Vaccinated (*n* = 424) [Pfizer = 332; Moderna = 92) delivered liveborn *n* = 85)	SM rate of 6.5%, preterm birth rate of 5.9%, FGR/SGA rate of 12.2%, cesarean delivery rate of 35.5%, congenital abnormality rate of 1.2%, stillbirth rate of 0%; NICU admission rate of 15.3%, any antenatal complication risk—23.5% All risks comparable to those of unvaccinated pregnancies
Zauche et al., 2021 [73]	Cohort study—USA	Vaccinated (*n* = 2456) [Pfizer =1294; Moderna 1162)]	Cumulative spontaneous miscarriage (SM) rate < 20 weeks (14.1 95 CI 12.1–16.1%) Risk of SM in standardised against maternal age for this population was 12.8%; 95 CI 10.8–14.8%), hence no difference in miscarriage rate/risk
Beharier et al., 2021 [42]	Multicentre cohort Israel	Pfizer Vaccinated—92; infected—74; control—66; reported delivery outcomes—92	Vaccinated versus infected versus controls, respectively Preterm birth—7.6% vs. 10.5% vs. 4.3% NICU admission—4.3% vs. 2.7% vs. 1.6%
Bleicher et al., 2021 [74]	Prospective cohort Israel	Pfizer Vaccinated—202 Unvaccinated—124	Outcomes presented in vaccinated vs. unvaccinated, respectively
Bookstein et al., 2021 [44]	Observational case-control Israel	Pfizer Vaccinated—57	Preterm birth—0% FGA/SGA—5.3% Hypertensive disorder of pregnancy—1.8% Caesarean section rate—17.6% Stillbirth/Neonatal death—0% NICU admission—3.5%
Dick et al., 2022 [75]	Retrospective—Israel	Vaccinated (Pfizer or Moderna)—2305 and unvaccinated—3313	Outcomes presented in vaccinated vs. unvaccinated, respectively SGA/FGR—6.2% vs. 7.050 Preterm birth—5.5% vs. 6.2% (NS)—vaccinated in 1st trimester Vaccinated in 2nd trimester - Preterm birth 8.1% vs. 6.2%, *p* < 0.001) Caesarean section—15.5% vs. 16.0% (NS) Stillbirth rate 0.87% vs. 1.0% (NS)
Goldshtein et al., 2022 [76]	Cohort—Israel	Vaccinated—7591, unvaccinated—16,697 Pfizer	Outcomes presented in vaccinated vs. unvaccinated, respectively The relative risk of fetal malformations—0.69 (CI 44–1.04) The relative risk of cardiac malformations—0.46 (CI 0.24—0.82) Preterm birth rate—4.4% vs. 4.1% (NS) FGA/SGA—6.6% vs. 6.7% (NS)
Kashani-Ligumsky et al., 2021 [45]	Cohort Israel	Pfizer Vaccinated—29 Infected—29 Control—21	Vaginal delivery—89.7% in vaccinated vs. 82.8% in infected vs. 85.7% in controls (*p* = 0.86)
Rottensteich M et al., 2022 [50]	Multicentre retrospective—Israel	Vaccinated with Pfizer vaccine—712; unvaccinated—1063	Outcomes presented in vaccinated vs. unvaccinated, respectively Preterm birth rate—1% vs. 0.9% (NS) FGR/SGA—11.4% vs. 9.2% (NS) Stillbirth rate—0.7% vs. 0.5% (NS) Caesarean section rate—15.%5 vs. 10.8% (*p* < 0.05) NICU admission—4.1% vs. 4.5% (NS) Composite adverse neonatal outcome—7.9% vs. 11.4% (*p* = 0.02) Composite adverse maternal outcome—24.2% vs. 23.6% (NS)
Wainstock et al., 2021 [77]	Retrospective cohort Israel	Pfizer Vaccinated—913 Unvaccinated—3438	Outcomes presented in vaccinated vs. unvaccinated, respectively FGA/SGA—2.8% vs. 3.8% (NS) Caesarean section—19.9% vs. 17.2% (NS)
Magnus et al., 2022 [78]	Case-control—Norway	Vaccinated (*n* = 1003) [Pfizer =790; Moderna = 137; AstraZeneca = 76] Unvaccinated—13,613	No difference in SM risk with an adjusted odd ratio of 0.81 for vaccinated versus unvaccinated for the previous 5 weeks
Magnus et al., 2022 [79]	Registry-based retrospective Norway and Sweden	Vaccinated—28,506; unvaccinated—129,015 Pfizer—20,424 Moderna—7607 AstraZeneca—475	Outcomes presented in vaccinated vs. unvaccinated, respectively FGA/SGA—7.8% vs. 8.5% (NS) Preterm birth—6.2/10,000 vs. 4.9/10,000 (NS) Stillbirth rate—2.1/100,000 vs. 2.4/100,000 (NS) NICU admission—8.5% vs. 8.5% (NS)
Fell et al., 2022 [80]	Retrospective Canada	Vaccinated—22,660 (Pfizer—18,101, Moderna—4507, Other—52) Vaccinated after pregnancy—44,815; Unvaccinated—30,115	Outcomes presented in vaccinated vs. unvaccinated vs. vaccinated after pregnancy, respectively Caesarean section—30.8% vs. 28.5 % vs. 32.2% (NS) NICU admission—11.0% vs. 12.8% vs. 13.3% (NS)
Sadarangani et al., 2022 [81]	Observational cohort Canada	Pregnant mRNA (Moderna and Pfizer) 5597—received 1st dose 3108—received 2nd dose Non-pregnant 174,765—received 1st dose 91,131—received 2nd dose Unvaccinated controls Pregnant—339 Not pregnant—5840	Outcomes presented in vaccinated vs. unvaccinated vs. vaccinated Miscarriage and stillbirth rate—1.5% vs. 2.1% (NS) Any adverse pregnancy outcome—0.9% vs. 0.6% (NS)
Blakeway et al., 2022 [82]	Retrospective cohort UK	Pfizer Vaccinated—133 Unvaccinated—399	Outcomes presented in vaccinated vs. unvaccinated, respectively Fetal abnormality—2.2% vs. 2.5% (NS) FGR/SGA—12% vs. 12% (NS) Caesarean section—30.8% vs. 34.1% (NS) NICU admission—5.3% vs. 5.0% (NS)
Citu et al., 2022 [56]	Prospective—Romania	Pfizer or Moderna Vaccinated (*n* = 173), unvaccinated (*n* = 529)	Outcomes presented in vaccinated vs. unvaccinated, respectively Preterm birth—8.1% vs. 6.9% (NS) Caesarean section—11.5% vs. 13.0% (NS)

**Table 4 viruses-15-00621-t004:** Recommendations and positions of the various societies on COVID-19 vaccination in pregnancy and reproduction.

Society	Recommend	Recommended Vaccine	Last Updated
American College of Obstetricians and Gynaecologists (ACOG)	Administration to all pre-pregnancy and throughout pregnancy and lactation	Moderna mRNA Vaccine Pfizer-BioNTech mRNA (preferred over J & J Janssen vaccine)	Dated April 2022
Society of Gynecology and Obstetrics of Canada (SGOC)	Administration to women during pregnancy in any trimester and while breastfeeding	All available COVID-19 vaccines approved in Canada can be used during pregnancy and breastfeeding. Presently, preference is given for the use of mRNA vaccinations during pregnancy as more data on safety and efficacy during pregnancy is available for these vaccines	14 March 2022
Royal Australian and New Zealand College of Obstetricians and Gynaecologists (RANZCOG)	Pregnant women in Australia and Aotearoa, New Zealand priority group for COVID-19 vaccination and should be routinely offered Women who are trying to become pregnant do not need to delay vaccination or avoid becoming pregnant after vaccination	Up to 3 doses of the Pfizer vaccine (Comirnaty) or Moderna (Spikevax) at any stage of pregnancy. Pfizer (Comirnaty) and Moderna (Spikevax) are mRNA vaccines. Global evidence has shown that the Pfizer and Moderna vaccines are safe for pregnant women. Novavax COVID-19 vaccine, a protein-based vaccine, can also be used in pregnancy (but is not currently approved for use in third or fourth doses). While there are no immunogenicity or safety data, there are no theoretical safety concerns relating to its use in pregnancy, since the Novavax COVID-19 vaccine, like other COVID-19 vaccines, is not a live vaccine	3 July 2022
International Federation of Gynecology and Obstetrics (FIGO)	Supports offering COVID-19 vaccination to pregnant and breastfeeding women	No specified vaccine	2 March 2021
National Health Service (NHS)	Administration to all pregnant and lactating women (considered high-risk group) and to those planning a pregnancy	Moderna mRNA Vaccine Pfizer-BioNTech mRNA preferred over J & J Janssen vaccine	March 2022
Royal College of Obstetricians and Gynaecologists (RCOG)	Strongly recommend administration to all pregnant and lactating women as well as those considering pregnancy	Comirnaty/Pfizer BioNTech or Moderna Spikevax mRNA vaccines, where available Women who had already had one dose of Oxford-AstraZeneca (before they became pregnant or earlier on in pregnancy) are advised to complete vaccination with a second dose of Oxford-AstraZeneca	15 December 2022
Society of Maternal Fetal Medicine (SMFM)	Administration to all unvaccinated individuals who are considering pregnancy, pregnant, recently pregnant, or lactating receive vaccination against COVID-19	Pfizer or Moderna bivalent mRNA COVID-19 vaccine	22 September 2022
Latin American Federation of Obstetrics and Gynecology Societies (FLASOG)	Vaccination against COVID-19 should be offered to pregnant or breastfeeding women	Not specified	18 February 2022
European Society of Human Reproduction and embryology (ESHRE)	Administration, if available to men and women attempting to conceive through assisted reproduction. Offer the COVID-19 vaccine before starting treatment or at any time during the fertility treatment or pregnancy	No particular vaccine preferred	25 February 2022
American Society of Reproductive Medicine (ASRM)	Administration to couples attempting pregnancy and during pregnancy ASRM Task Force recommends that physicians should follow CDC guidance	mRNA-preferred, but people aged 18–49 who received J & J Janssen COVID-19 vaccine and booster who are not moderately or severely immunocompromised should receive a second booster dose using an mRNA COVID-19 vaccine at least 4 months after the first Janssen booster dose	November 2021
European Board and College of Obstetrics and Gynaecology (EBCOG)	The possibility of vaccination should be offered to all pregnant women after being adequately informed of the benefits and risks. EBCOG urges all Health Authorities and Governments to make vaccination available to all pregnant women wishing to take them. EBCOG supports that COVID-19 vaccination be recommended to all breastfeeding women in the absence of a specific contraindication	Vaccine type not specified	14 May 2021
Center for Disease Prevention and Control (CDC-USA)	Administration to pregnant, lactating and those trying to conceive	mRNA and J & J Janssen vaccines	April 2022

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
