# Peer review of "Immunisation against COVID-19 in Pregnancy and of Women Planning Pregnancy"

_viruses, 2023, doi:10.3390/v15030621_

Round 1
Author Response
We have responded to all the queries and suggestions made by the referee. These are included in the attached file

Reviewer 2 Report
A well and impactful review/commentary journal to discuss COVID-19 immunisation issue in reproduction and pregnancy. The urgently approved vaccines are effective to eradicate severe infection by SARS-CoV-2. Still, public concerns exist to whether the vaccines are most safe. The current article provides evidences and references to support the fact that adverse effect of the existing approved SARS-CoV-2 vaccines are low at pregnancy and reproduction. Even adverse effect could still happen, selecting a correct vaccine can work (e.g. avoid adenoviral vector type during pregnancy to prevent prothrombosis). The content is informative. I would like to suggest for some minor corrections that could improve the article for publication. The comments are as follow:
1. For line 36, “Coronaviruses cause various infections in mammals including birds and human” need to be rewritten. Sentence might be misleading for mammals and birds.
2. For line 42, underestimation to death rate might not be accurate. In fact, asymptomatic infection exists that also lends to unknown true infection rate.
3. For line 45-46, “This family of viruses have a large RNA genome and therefore a propensity for genetic variation” might not be accurate. SARS-CoV-2 express its proofreading enzyme that corrects genetic error that is however lower than other RNA virus such as influenza.
4. reference 9 and 10 are duplicated.
5. Line 253, the number “178” is error.
6. The format of the reference list is not organised well. The format is inconsistent and needed for revision.
7. For line 208-209, what do you mean by other studies? Could you provide with reference?
8. Could you explain in brief for why pregnancy will have double the risk for VTE (line 207)
Author Response
The response to reviewer 2 and 3 are included in the uploaded file

Reviewer 3 Report
The review is well conducted and written very clearly. The data is up to date and well interpreted. I would make a single consideration regarding the absence of tables that could make the paper even more attractive (just as an example, a table showing the rate of complicated pregnancies from the V-safe pregnancy registry (without and with vaccine) and a table with the recommendations of the authors (in line with those of the CDC, ECDC, WHO) correctly expressed in the text (e.g. recommended type of vaccine (mRNA, viral vector, etc), timing of administration etc. I believe the topic "vaccines and breastfeeding" is poorly represented which, although not included in the title of the paper, is certainly of great general interest after pregnancy. I would therefore integrate it with a short, specific paragraph. Congratulations on your work.
Author Response
Already submitted with reviewer 2

Round 2
Reviewer 1 Report
Dear Authors,
Your manuscript have been re-reviewed.
I would like to thank you for all the modifications you did.
In its present form your article is better than the first version, but I have the following remarks to be taken into consideration:
01- The abstract needs paraphrasing.
02- Authors affiliations' numbers must be after the authors' names.
03- Your text must be justified.
Best Regards,
Author Response
The file contains all the responses to referee no 1
